# A Novel Steganography Method for Character-Level Text Image Based on Adversarial Attacks

**DOI:** 10.3390/s22176497

**Published:** 2022-08-29

**Authors:** Kangyi Ding, Teng Hu, Weina Niu, Xiaolei Liu, Junpeng He, Mingyong Yin, Xiaosong Zhang

**Affiliations:** 1Institute for Cyber Security, School of Computer Science and Engineering, University of Electronic Science and Technology of China (UESTC), Chengdu 611731, China; 2Institute of Computer Application, China Academy of Engineering Physics, Mianyang 621900, China

**Keywords:** steganography, adversarial attack, transferability, OCR models

## Abstract

The Internet has become the main channel of information communication, which contains a large amount of secret information. Although network communication provides a convenient channel for human communication, there is also a risk of information leakage. Traditional image steganography algorithms use manually crafted steganographic algorithms or custom models for steganography, while our approach uses ordinary OCR models for information embedding and extraction. Even if our OCR models for steganography are intercepted, it is difficult to find their relevance to steganography. We propose a novel steganography method for character-level text images based on adversarial attacks. We exploit the complexity and uniqueness of neural network boundaries and use neural networks as a tool for information embedding and extraction. We use an adversarial attack to embed the steganographic information into the character region of the image. To avoid detection by other OCR models, we optimize the generation of the adversarial samples and use a verification model to filter the generated steganographic images, which, in turn, ensures that the embedded information can only be recognized by our local model. The decoupling experiments show that the strategies we adopt to weaken the transferability can reduce the possibility of other OCR models recognizing the embedded information while ensuring the success rate of information embedding. Meanwhile, the perturbations we add to embed the information are acceptable. Finally, we explored the impact of different parameters on the algorithm with the potential of our steganography algorithm through parameter selection experiments. We also verify the effectiveness of our validation model to select the best steganographic images. The experiments show that our algorithm can achieve a 100% information embedding rate and more than 95% steganography success rate under the set condition of 3 samples per group. In addition, our embedded information can be hardly detected by other OCR models.

UTF8gkai

## 1. Introduction

With the great epidemic of COVID-19, the way entities communicate with each other is increasingly moving online. They need to ensure that their data are not compromised, especially important confidential data. However, with today’s Internet, attackers can intercept transmissions by a variety of techniques, so protecting transmissions remains a necessity to ensure information security. Steganography is an important method of confidential communication, and its carrier can be propagated over open channels due to its covert embedding, which has been a hot research topic in data protection. The carrier of steganography, i.e., the form of the covered data, can be texts, images, audio, or videos. At the embedding end, steganography algorithmically embeds the confidential data into the redundant space of the public carrier data to generate the stego data. At the receiving end, the receiver separates the confidential data from the cover data. The goal of steganography is to embed the specified information in the corresponding carrier, and the embedded information is not visible to third parties. In addition, the receiver can effectively extract the embedded information, and non-specified receivers cannot extract the embedded information.

The existing image steganography methods mainly contain modifying LSB bits and LSBR to embed information [1], steganography based on statistical property preservation [2,3], content adaptation steganography [4,5], adaptive steganography based on minimizing distortion [6,7,8,9], and steganography based on deep learning [10,11,12,13,14,15,16]. Existing steganography methods require custom algorithms or models to implement information embedding and extraction. Manually customized information extraction methods may be deciphered, and the models used for decryption may be subject to forensics by the detecting party.

Our approach is mainly oriented to a subfield of image steganography, scanned (photographed) Chinese text image steganography. Firstly, human language is denser with information than natural images, and thus text images can contain more information than ordinary images. Secondly, scanned text images have more room for adding perturbations and are more tolerant to noise than computer output text images. The text image steganography of computer output is closer to text steganography, and the redundancy space it can be embedded is very limited. The Chinese language was chosen because, compared to English, the Chinese language uses characters of similar width to correspond to the English word as the basic unit. The inconsistent length of English words can add restrictions to the steganographic embedding. Since the number of characters in English words varies, if we perform character replacement for a word, we need to find the word with the number of characters equal to the number of characters in the content to be replaced for replacement, which lacks flexibility compared to the replacement of Chinese characters. Therefore, our method is more suitable for steganography of character unit characters like Chinese in terms of working principles.

Compared to existing methods that require manual customization rules or training custom models to achieve information embedding and extraction, our approach uses ordinary OCR models to achieve steganography. The OCR models we use are difficult to associate with steganography even if they are intercepted. This ensures the security of our steganography method. In addition, due to the random nature of the training of AI models, a third party cannot reproduce an identical model by the same training means. Our approach exploits the vulnerability of AI models by performing an adversarial attack on text images in the spatial domain to change the recognition results of AI models under the condition that the human eye recognition results are hardly changed. In addition, for text image steganography applications, we have proposed a corresponding adversarial transferability weakening method so that the adversarial attack is effective only for our local model. The scenario for the application of our approach is that the sender and the receiver need to store 2 identical OCR models on their own devices. These two models, one for embedding and extracting information and the other for verifying the extracted information, enable fast extraction of information. The models we use are not different from the ordinary OCR models, perform the OCR tasks normally, and are not specifically designed for steganography. Therefore, we consider it safe to store these two OCR models on their devices. The schematic diagram of our steganography method is shown in Figure 1.

Our contribution is as follows:We propose a character-level steganography method for text images. The method uses an adversarial attack to embed secret information into text images, such that the local OCR model recognizes the text image as the wrong result, and the wrong result is the information we embed. Our method does not have manually customized steganography rules or algorithms, nor does it train a model for steganography, but uses a normal OCR model, which makes the embedding and extraction of information more covert and has forensic resistance, all of which increases the security of our proposed steganography method.To reduce the perturbation required to embed the hidden data, we combined the characteristics of text images and select the text region with the greatest similarity (not identical) to embed information. Experiments show that this strategy of ours substantially improves the success rate of information embedding while reducing the perturbation required to embed the information.We weakened the adversarial transferability by placing the generated steganographic images at the decision boundary of the local model, thus avoiding the detection of our embedded information by other OCR systems. At the same time, we used a validation model to acquire the low transferability steganographic images for the optimal output screening, further weakening the adversarial transferability of the generated images. Decoupling experiments and parameter selection experiments show that our method reduces the possibility of generating steganographic images to be detected by other OCR models and achieved a better steganography success rate.

This paper is structured as follows. In Section 1, we introduce the background of the application of our method and our main contributions; in Section 2, we introduce the recent research related to adversarial samples and the applications of deep learning to steganography; in Section 3, we introduce our background; in Section 4, we introduce our method; in Section 5, we introduce the experimental setting of our method and analyze and discuss the experimental results; in Section 6, we show the implications of this paper; in Section 7, we conclude our paper.

## 2. Related Work

### 2.1. Adversarial Samples

The security of artificial intelligence algorithms represented by deep learning has received much attention in recent years. Since the discovery of adversarial samples by Szegedy [17], researchers have conducted in-depth research on adversarial samples [18,19,20,21,22,23,24]. The adversarial sample generation process for artificial intelligence models is shown in Figure 2.

Researchers have conducted some studies on the generation of OCR adversarial samples. In general, character-level attacks on English OCR models are less difficult than those on Chinese OCR systems because English has fewer elements and a more similar structure; whereas Chinese characters have a very large number of constituent elements and a significantly different structure, so attacks on Chinese OCR systems are usually difficult to avoid perturbations visible to naked eyes. Ref. [25] uses the objective function of [19] to implement the adversarial attack, and its work is mainly focused on maintaining the correctness and consistency of the semantics. Ref. [26] has implemented the adversarial attack on Chinese character OCR systems by adding a combination of adversarial samples and visible watermarks, but its success rate against Chinese characters is still low compared to other domains.

According to the degree of knowledge about the attacked model, adversarial attacks are classified into black-box and white-box attacks. The white-box attacks mean that the attackers have complete information about the parameters and structure of the model, and the attackers can generate the adversarial samples by gradient calculation. The black-box attacks mean that the attackers do not have any information about the parameters and structure of the victim model, and can only generate the adversarial samples by inferring the input and output of the victim model.

Transfer-based adversarial attacks are an important black-box attack method. Transfer-based black-box attacks refer to an attacker generating adversarial samples (white-box attack) by attacking local models that he has grasped and using the generated adversarial samples to attack a victim model that he does not grasp. To improve the attack success rate of transfer-based attacks, researchers have proposed several methods [20,27,28,29,30]. At present, almost all the research on adversarial sample transferability focused on enhancing the transferability of adversarial samples to perform adversarial attacks. However, in steganography applications, the existence of adversarial transferability increases the probability that we will be detected by the detectors using other OCR models, and, therefore, we need to reduce the transferability of the adversarial attack.

### 2.2. Deep Learning in Image Steganography

To avoid detection by steganography detectors, researchers have proposed image steganography techniques based on generative adversarial networks [31] and adversarial attacks.

The image steganography methods based on generative adversarial networks consist of three main approaches. The first one is to generate carrier images suitable for embedding information and then embed information using traditional steganography methods [10,11]. The second one is the embedding carrier-based steganography, which embeds the hidden information on the natural image using generative adversarial networks [32,33,34]. The third one is synthetic carrier-based steganography, which synthesizes the original carrier with the hidden information to generate a new semantic image containing the hidden information [35,36]. The generators and detectors of GAN-based steganographic methods of steganography are confronted with each other in the training process and performance gains are obtained in the confrontation.

The adversarial attack-based image steganography methods use adversarial attacks to yield erroneous output from the steganalyzer while ensuring that the steganographic information is conveyed [15,16]. Unlike GAN-based detectors that are continuously optimized during training, the steganography approach based on adversarial attacks generates steganographic data with the goal of avoiding detection by existing steganographic detectors. However, this approach is limited by the ability of adversarial sample transferability, and its steganographic performance is degraded under the analysis of unknown steganalyzers.

Although existing deep learning-based image steganography methods require the custom model for embedding and extracting hidden information, we exploit the complexity and uniqueness of neural network boundaries and use the ordinary OCR models as a tool for information embedding and extraction. At the same time, we reduce the transferability of the adversarial attack so that our embedded information can only be recognized by our local model, even if models trained by different batches using the same training data and the same structure are difficult to recognize. In addition, we modify only individual characters, controlling the amount of perturbation that needs to be added.

## 3. Background

At present, deep learning has become the mainstream method for OCR tasks, and this paper uses a Convolutional Recurrent Neural Network (CRNN) [37] as the local model. CRNN is an end-to-end OCR task model, which uses CNN to extract convolutional features of text images, and then uses LSTM to sequentially analyze the features extracted by CNN. Finally, the model is trained by using the Connectionist Temporal Classification (CTC) [38] loss function, which solves the problem of difficult character alignment during training.

The CTC loss function is a loss function for a wide range of NLP applications, including OCR model training. The CTC loss function provides an alignment-free labeling method for end-to-end sequential neural network training. Suppose *x* is a input of the OCR model, π is the output of the model f(x), $f\left(x\right)=\pi =\left[\pi^{1},\pi^{2}\cdots \pi^{n}\right]$, the label of input *x* is $L=[L^{1},L^{2}\cdots L^{m}]$, where m<n. This method calculates P(L|x) by exhausting every possible combination with output *L* and calculating its probability. The set whose output is *L* is defined as $S=[s^{1},s^{2}\cdot s^{n}]$. P(L|x) s calculated, as shown in Equation (Equation 1).
(1)$$p\left(L|x\right)=\sum_{L^{i}\in S}\prod_{i=1,n}p\left(s^{i}|x\right)=\sum_{s^{i}\in S}\prod_{i=1,n}{\left(\pi^{i}\right)}_{s^{i}}$$

The loss function is calculated by the input of *x* and the label of *L* as shown in Equation (Equation 2).
(2)$$L_{\mathrm{CTC}}\left(f\left(x\right),L\right)=-\log p\left(L|x\right)$$

## 4. Methodology

The character-level text image steganography method based on adversarial attacks is mainly divided into the information embedding phase and the information extraction phase. In the information embedding phase, we use the adversarial attack to embed the information into the text image in the form of a perturbation that makes the local model generate errors. In the extraction phase, we restore the embedded information by capturing the results of the local model recognition errors. The framework of the character-level text image steganography method based on adversarial attacks is shown in Figure 3.

### 4.1. Information Embedding

To implement the character-level image steganography method based on adversarial attacks, we first pre-process the images to ensure that both the local model and the reference model can correctly recognize the image information. Next, we measure the similarity between the carrier characters and the characters to be embedded, and select the most similar carrier character for modification; then, we implement the adversarial attack on the chosen carrier characters, and we weaken the adversarial transferability by keeping the generated embedded characters at the decision boundary of the local model; finally, to further reduce the probability of being recognized by other OCR models, we use an independent validation model to filter the generated embedded data.

#### 4.1.1. Text Image Pre-Processing

When extracting the embedding information, we are extracting the result that makes the local model recognize it incorrectly. Therefore, we need to ensure that the local model *M* recognizes all our initial text images correctly. Additionally, to improve the extraction efficiency and to avoid passing the labels of the text images, we use a reference model *R* to quickly extract the text labels (no manual checking of information is required), which serves to recognize our transmitted text images with a 100% correct rate. Therefore, the goal of the text image pre-processing is to allow the local model and the reference model to correctly recognize the all characters in the text image. In case of incorrect recognition, an adversarial attack is applied to the text image to ensure that the local model and the reference model can recognize the input text image with 100%. The target of the attack is then set to the label of the correct result, and the loss function we use is the CTC loss function. When $D\left(M\left(x\right)\right)\neq L_{ori}$, our preprocessing is to optimize the perturbation δ satisfying:(3)$$\begin{array}{c} \mathrm{minimize}\;L_{\mathrm{CTC}}\left(M\left(x+\delta \right),L_{ori}\right)+L_{\mathrm{CTC}}\left(R\left(x+\delta \right),L_{ori}\right)\\ \mathrm{such}\;\mathrm{that}\;x+\delta \in {\left[0,1\right]}^{n} \end{array}$$

$L_{ori}$ denotes the ground-truth label of the input *x*, D(·) removes blanks and sequential duplicate characters.

#### 4.1.2. Similarity Measure

To solve the problem of adding too much perturbation to implement adversarial attacks on the OCR system of Chinese characters, we selected the location of the embedded characters. We use the CTC loss function to measure the similarity between the carrier characters and the embedded characters.

Since the text is serialized structure. To parallelize the text embedding using adversarial attacks, each group is fixed to correspond to only one character that needs to be embedded, and each group contains at least one sample (the sample is an image input to the OCR model). Therefore we modify one label at a time and compute the CTC loss function for that sample input to the local model with the embedded label. For example, the original label is $L=[L^{1},L^{2},\cdots L^{m}]$, and the label after embedding the information at the first character is $E1=[e,L^{2},\cdots L^{m}]$, where *e* denotes the character we need to embed in this sample. Next, $\left[E^{1},E^{2},\cdots E^{m}\right]$ is sorted to obtain the label corresponding to the minimum CTC loss function value. Finally, we use the label with the smallest CTC loss function in each sample to perform the adversarial attack. If the embedded character is the same as one of the characters in the carrier sample, we do not use that embedding position.

#### 4.1.3. Information Embedding Based on the Low Adversarial Transferability Attack

In general, when the targeted attack is successful, the larger the distance of the adversarial sample from the decision boundary, the stronger the adversarial transferability. To reduce the transferability of the adversarial sample, we want the image after embedding the information to be located just near the decision boundary. To achieve this goal, we add a boundary control coefficient con for controlling the distance relationship between the embedded samples and the decision boundary. We design a simple control coefficient strategy as shown in Equation (Equation 4).
(4)$$con=\left\{\begin{array}{c} 0\;\mathrm{if}\;\;D\left(M\left(x\right)\right)=E_{min}\\ 1\;\mathrm{else} \end{array}\right.$$

D(·) removes blanks and sequential duplicate characters, and $E_{min}$ denotes the minima CTC similarity labels between original images and embedded information.

To reduce the visibility of the added perturbation, we used the $L_{2}$ parametrization as a regular term, where $c_{2}$ is the coefficient of the regular term. In addition, to ensure the correct identification by the reference model, we also add the requirement for the reference model output in the information embedding process. $c_{1}$ is the coefficient of the reference model CTC loss function. Our information embedding objective function is shown in Equation (Equation 5).
(5)$$\mathrm{minimize}\;(con\cdot L_{CTC}\left(M\left(x+\delta \right),E_{min}\right)+c_{1}\cdot L_{CTC}\left(R\left(x+\delta \right),L\right)+c_{2}\cdot \left||\delta \right|{|}_{2})$$

We used the momentum method as an optimizer to optimize our objective function for smoothing the control coefficient con. We want the optimization process to proceed along the model decision boundary surface. The con coefficient is updated in real-time for each iteration step, while the momentum coefficient μ facilitates the optimization in the original direction, thus reducing the impact due to the con coefficient mutations and making the whole optimization process more stable. The optimization process is shown in Equation (Equation 6).
(6)$$\begin{array}{cc} g_{0}= & 0\\ g_{t}= & \mu g_{t-1}+\nabla_{x}loss\left(x^{\left(t-1\right)}\right)\\ x^{\left(t\right)}= & x^{\left(t-1\right)}+\alpha \cdot g_{t} \end{array}$$loss denotes the objective function (Equation 5). $g_{i}$ denotes the gradient at the *i*-th iteration after optimization. α denotes the learning rate.

### 4.2. Sample Filtering

To further improve the success rate of embedding information and reduce the recognition rate of other OCR models, we expand the sample size of each group, and the task of each group remains to embed one character. For this purpose, we introduce a new validation model *V*, which is trained in the same way, with the same model structure, and using the same training data as the local model M. The role of the validation model *V* is to filter out the data with the least impact of the added perturbation on the output of other OCR models for the embedded information, i.e., the correct label is still output with a high probability by the validation model. The filtering is completed by calculating the CTC loss function of the output of the embedded image input model *V* generated by each group of samples with respect to the original label. Finally, we filter out the samples with the smallest CTC loss function and successful embedding. To meet the requirement, we use a modified CTC loss function. As shown in Equation (Equation 7).
(7)$$L_{V}=\frac{L_{CTC}\left(V\left(x\right),L\right)}{succ_{M}+1e\times 10^{-8}}$$

$succ_{M}=1$ while $D\left(M\left(x\right)\right)=E_{min}$, $succ_{M}=0$ while $D\left(M\left(x\right)\right)\neq E_{min}$. At $succ_{M}=0$, the value of the loss function will increase significantly. This ensures that the samples filtered by our method ensure that the local model outputs the specified results.

Assuming the number of samples in each group is 3, our algorithm is shown in Algorithm 1.
**Algorithm 1:** The character-level image steganography method
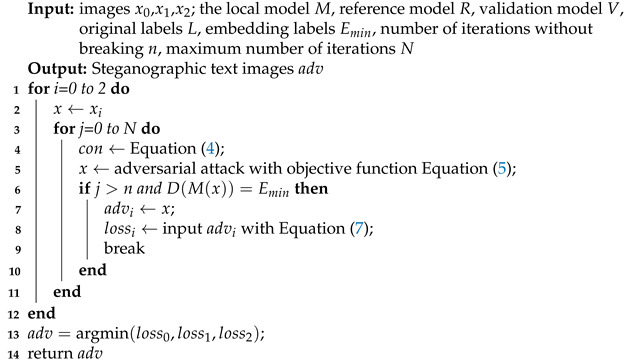


We use the number of termination-free iterations *n* because, according to the literature [39], directions close to the initial gradient direction have stronger adversarial transferability, and we want to use more iterations in the optimization process, as far away from the initial gradient direction as possible.

### 4.3. Extraction of Embedded Information

Our steganography scenario assumes that the message sender and receiver hold a copy of the local model and the reference model offline before the messaging requirement occurs and that they are saved separately.

The information we embed is only recognized by the local model while leaving the output of the other models still with the original results. To simplify the extraction process and prevent the transmission of label information, we use a dual model authentication system. In which the original label information is provided by the reference model R, while the embedded information is the different results provided by the local model and the reference model. The formula for extracting the embedded information is shown in Equation (Equation 8).
(8)message=D(M(x)×D(M(x)⊕D(R(x))))

⊕ denotes the exclusive OR symbol.

## 5. Experiment Results

In this section, we first introduce the setup of our experiments; then, we present the evaluation metrics for the experiments; next, we perform decoupling experiments on our steganography algorithm to verify the effectiveness of similarity sorting and ensuring the position of the generated images in relation to the boundary; then, we demonstrate the effectiveness of using the validation model by testing the performance of our algorithm under different hyperparameter settings; at last, we analyze the performance of our algorithm.

### 5.1. Experiment Setting

The Chinese dataset we chose is Synthetic Chinese String Dataset [40], which is derived from Chinese news, literature, and literary texts, and is generated by size, grayscale, clarity, perspective, and stretching operations. The dataset contains a total of 5990 characters, including Chinese, punctuation, English, and numbers. Each sample contains 10 characters with a resolution of 280×32, and the total number of samples is about 3.6 million.

Due to a large number of characters in Chinese characters, to avoid using characters that are not included in the Synthetic Chinese String Dataset and evaluate the information embedding capability of the algorithm objectively, the characters we embed are selected from the commonly used characters in Chinese characters, and the statistics of the commonly used characters list are from the literature [41]. According to the statistics, the top 2000 high-frequency used Chinese characters cover more than 95% of the usage scenarios. Therefore, to take into account both the textual capacity of the dataset and the normal semantic communication, we embed the information for randomly selecting the top 2000 high-frequency Chinese characters.

Networks with the same structure in the same training environment have more similar decision boundaries. When OCR models with the same structure are difficult to check the embedding information, OCR models with other structures are more difficult to identify. Because OCR models with the same structure have more similar decision boundaries, and the difference in their decision boundaries is entirely due to the difference in initialization and the randomness in the training process; other models will have a greater difference in the similarity of their decision boundaries due to the dissimilar structure and the randomness in the initialization and training process. The greater the decision boundary similarity, the more likely it is to be the same as the decision result of the local model after adding the required perturbation for steganography; on the contrary, the greater the difference in the decision result. If the success rate of the same structural OCR model to recognize our embedded information is very low, thus there is no need to show the extraction success rate of models with large structural differences. In our experiments, we use the same architecture and training method for the local, reference, and validation models, and the architecture of the network is CRNN. The CRNN we use contains 6 convolutional layers, 4 pooling layers, and, finally, the output of the convolutional layers is fed into the Bi-LSTM network [42]. Each model was trained in 50 steps using the CTC loss function. 99.03%, 98.93%, and 99.05% accuracy were achieved on the test set for the local, reference, and validation models, respectively. For testing, we used a completely new testing model for testing the embedded information, which was trained in the same way as the local model.

We set the information embedding as follows, the reference model coefficient $c_{1}$ is 0.2, and the regular term coefficient $c_{2}$ we set to 0.01. The number of the initial optimization iterations *n* is 1600, the number of the maximum optimization iterations is 2000, and the momentum coefficient μ is 0.2. In the testing process, we randomly selected 100 characters from the top 2000 high-frequency words for embedding each time, and 10 sets of experiments were performed for each experiment, while the embedded images were randomly selected from the test dataset.

Our experiments were executed on a server with an Intel(R) Xeon(R) Gold 6248R CPU @ 3.00 GHz CPU, 128 GB system memory, 2 GeForce RTX 2080Ti, and Pytorch version is 1.7.0.

### 5.2. Metrics

We evaluate the algorithm’s ability to embed information in terms of three dimensions: the ability to extract the added information, the difficulty to be recognized by other OCR models, and the size of the added perturbation, respectively. Based on these 3 requirements, we have proposed the following 5 metrics.

**Information extraction success rate**: we use the local model to extract the embedded information of text images. Since both information embedding and information extraction are performed by using the local model, the information embedding success rate is equal to the success rate of information extraction.**Correct recognition rate of testing OCR models**: invisibility to other OCR models is very important for information hiding, and our algorithm needs to ensure that other models can still correctly recognize the original information of our steganographic images.**Probability of no impact on testing OCR models’ results**: if only the recognition correctness of other OCR models is the goal, the detector can use other OCR models to detect the output of text images and obtain the embedded information if the detection results of a large number of characters are at the boundary point. To meet this challenge, we require that the increase in the CTC loss function of the image after embedding the information in the recognition results of other OCR models must be less than a threshold value th to achieve consistency in the CTC loss function and avoid detectors using other models to recover the embedded information. The condition that needs to be satisfied to have no impact on other OCR models is shown in Equation (Equation 9).
(9)$$L_{CTC}\left(V\left(x\right),E_{min}\right)-L_{CTC}\left(V\left(x\right),L\right)<th$$In our experiments, th was set to 0.01.**Steganography success rate**: we consider that the definition of steganography success is not only that the embedded information can be extracted (metric 1), but also that the embedded information has no impact on other OCR models (metric 3). Steganography success rate contains two requirements, the first is that the steganographic information is correctly extracted in the steganographic image, and the other is that other OCR models cannot access the hidden information, here we use the higher standard that the added perturbations do not have an impact on other OCR recognition results. The success rate of steganography is calculated by the Equation (Equation 10).
(10)P(S)=P(E)×P(O)
where P(S) denotes the success rate of steganography, P(E) denotes the success rate of information extraction, and P(O) denotes the probability of no impact on other OCR models.**Perturbation size**: adding too much perturbation will cause it to be easily detected by the human eye or other detection algorithms, and we use the $L_{2}$ distortion to measure the size of the perturbation.
(11)$$L_{2}=\frac{1}{N}\sum_{i=1}^{N}\left|\right|\delta_{i}{\left|\right|}_{2}$$*N* denotes the number of groups.

### 5.3. Decoupling Experiments

The purpose of our decoupling experiments is to verify the effectiveness of our adopted strategy in improving steganographic capabilities. We tested the performance of our approach when we do not employ similarity sorting of the embedded characters to the original carrier and limiting the sample-boundary location relationship during optimization, respectively. To simplify the description, we refer to the similarity ranking as strategy ① and the sample-boundary position relationship qualification during optimization as strategy ②. In this experiment, we set the number of samples per group to 1, and all experiments do not use the validation model. The statistical figure of extraction success rate is shown in Figure 4, and the statistical figure of no impact on the recognition results of other OCR models is shown in Figure 5.

As shown in Figure 4, the distribution of the extraction success rate of each of these methods is focused. Except for the method that does not use ①, all other methods can extract information with more than a 90% success rate, and only the method that uses ② is lower than the other three methods by about 30%. Therefore, it can be obtained that strategy ① has the greatest influence on the information extraction success rate. The embedding success rate of the method without strategy ② is higher than that of the method with ① and ②. It can be concluded that the use of strategy ② causes a small decrease in the embedding success rate because strategy ② reduces the perturbations that drive local model decisions as embedded information to decide as a result of steganography.

As shown in Figure 5, the probability of no impact on the recognition results of other OCR models is significantly lower than the success rate of information extraction, and both strategy ① and strategy ② can reduce the impact on the recognition results of other OCR models with a similar degree. In addition, strategies ① and ② used together can further reduce the impact on the recognition results of other OCR models.

The averages of the success rate of information extraction, the correct recognition rate of other OCR models, the probability of no impact on the recognition results of other OCR models, the success rate of steganography, and the perturbation size are shown in Table 1.

The probability of no impact on other OCR models is much lower than the probability when the recognition result is unchanged for other OCR models, indicating that the former is the higher standard. For the steganographic success rate, the best steganographic success rate is obtained by the method using strategies ① and ②; the methods without using strategies ① and ② have similar steganographic success rates as the method without ①, mainly due to the lower success rate of information extraction without using ①. In terms of perturbation, the method using both strategies ① and ② has the smallest steganographic success perturbation, and both strategies ① and ② have the effect of reducing the steganographic perturbation by a similar amount.

The steganographic images generated by our method are shown in Figure 6.

As can be seen from Figure 6, the perturbations added by our method to embed the information are difficult for the human eye to capture.

As shown above, we can obtain that strategy ① can reduce the perturbation required for steganography by significantly reducing the difficulty of information extraction, which, in turn, reduces the probability of impact on the recognition results of other OCR models; method ②, although it reduces the success rate of information embedding, can significantly reduce the probability of impact on the recognition results of other OCR models and reduce the required perturbation.

Decoupling experiments show that the strategy we use improves the success rate of information embedding, reduces the probability of being recognized by other models, and also reduces the perturbations required for steganography.

### 5.4. Hyperparameter Selection

In this part, we explore the relationship between the parameters in the steganography algorithm and the steganography functions, including the relationship between the regular term coefficients and the steganography functions, and the relationship between the number of samples in each group and the steganography functions.

#### 5.4.1. Regular Term Coefficient

The regular term coefficient is used to balance the steganographic success rate with the size of the generated perturbation. The number of samples per group is set to 1. The results are shown in Table 2.

As shown in Table 2, the experimental results are following our expectations. With the increase in the regular term coefficients, the success rates of embedding information of the local model are decreasing, and the recognition success rates of other models are increasing. The steganography success rates show a trend of increasing and then decreasing, while the perturbation is decreasing. A lower perturbation size means a smaller probability of being detected, therefore, we consider a regular term coefficient of 40 as the optimal regular term coefficient for our algorithm.

#### 5.4.2. Number of Samples per Group

We increased the steganographic success rate of our steganography method by increasing the number of samples per group, but this leads to a decrease in the amount of information embedded. For this experiment, we used a validation model to filter the generated steganographic images when the number of samples per group is larger than 1, which is used to reduce the impact on other OCR models. The evaluation of the steganography performance for the different number of samples per group is shown in Table 3.

As Table 3 demonstrates, our steganography success rate increases as the number of samples per group increases. When the number of samples per group is greater than 2, the success rate of information embedding by the local model exceeds 99%. When we used the validation model for filtering, the probability of our steganographic text images being recognized by other models as original information increased dramatically. Therefore, the filtering of steganographic images using the validation model is effective. The probability of the model being detected by other models is less than 5% when using samples larger than 2 per group. If using the hard labels for judgment, the probability of our steganographic images being recognized by other OCR models is less than 1.5% when the model is larger than 2 (lower than the model’s false recognition rate for normal text images). In terms of perturbation, there is no significant trend change in the magnitude of perturbation as each group of samples increases. We believe that the perturbation variation in this experiment is more related to the difficulty of information embedding using the original images.

Hyperparameter selection experiments find the optimal parameters for our steganography algorithm, and we also verify the effectiveness of the validation model filtering we use.

### 5.5. Algorithm Performance Analysis

We analyze the performance of the algorithm in terms of its capability, security, payload, and robustness.

**Capability**: Table 4 shows the confusion matrix of embedded and extracted information for each group of only one sample of our algorithm. Positive represents the number of characters with embedded information and negative represents the number of characters without embedded information; true is the number of extracted characters with the same embedded information, false is the number of extracted information with different embedded information, and none is the number of failed extracted information. It can be seen that most of the cases of extraction failure are failure to extract information, i.e., the output of the local model is the original character information, and a small number of errors occur in the case of extraction of other information. In order to avoid extracting wrong information, we need to overwrite the current modified sample with the pre-processed sample if the embedding fails in the information embedding phase. As shown in Table 3, our algorithm achieves a 100% extraction success rate when the number of samples per group is greater than or equal to 3. It proves the capability of our algorithm for information embedding and extraction.

**Security**: Since the way our algorithm embeds information is different from previous steganography methods and the purpose of the added perturbations is completely different, it is difficult for traditional statistics-based steganography analyzers to perform steganography analysis on our algorithm. We input the original image and the steganographic image into the deep learning-based steganographic analyzer SRNet [43], whose detection rate confusion matrix is shown in Table 5. It can be seen that SRNet is completely unable to recognize our steganographic images. We believe that it is the other OCR models that are the most realistic risk to our approach. As shown in Table 3, the recognition rate of the validation model with the same structure as the local model for embedded information is 1.2% for each group of samples greater than or equal to 3, while the probability of having an effect on the output of the testing model is 4.1%. The larger the number of samples per group, the smaller the probability that our embedding information is perceived. As shown in Figure 6, the perturbations added by our steganography are invisible to the human eye. In addition, unlike other steganography methods that have manually customized extraction algorithms or information extraction models based on machine learning, our stored OCR models are not customized for steganography and have better hiding properties. This will help improve the resistance of our steganography methods to physical forensics.

**Payload Capacity**: In the experiment, each sample contains 10 characters. As shown in Table 3, we consider an information extraction rate of 100% to be acceptable. Based on the way our steganographic images are generated, we define our algorithm as a payload capacity of 1 character for every 30 characters. Our payload capacity is significantly different from the normal steganography algorithm. In contrast, since the continuous sending of text images is a relatively normal event, it is entirely possible to achieve high-capacity information embedding by continuously sending the associated scanned document images. However, the payload capacity may still limit the scope of application of our algorithm.

**Robustness**: Since the information of our algorithm is embedded near the decision boundary of the local model, our algorithm is almost not robust to noise. We have used JPEG compression to compress our steganographic images and reused our extraction method for extraction. Table 6 shows the success rate of steganographic information extraction by the local model for steganographic characters and the correct recognition rate for other characters, as well as the recognition correct rate of the reference model for all characters with JPEG compression quality.

As Table 6 shows, the extraction success rate decreases very significantly with compression quality. The local model recognizes other characters with less than 100% accuracy, while the reference model recognizes all characters with less than 100% correctness. This all leads to the embedded information not being extracted. However, the model’s sensitivity to other perturbations can be used to detect whether a steganographic image has been tampered with by others. In case of tampering, a completely new communication path is used for transmission.

Our algorithm performs well in terms of capability and security, while in terms of robustness, although we can detect whether a third party has tampered, it still limits the practical application of our algorithm.

## 6. Implication

We propose a new steganography method whose carrier is a text image. The method exploits the stochasticity of the decision boundary of the AI model to embed the information by weakening the transferability of the adversarial attack. The advantage of this method is that it does not use specific algorithms or specially trained steganographic models for embedding and extraction of information. This work improves the concealment of steganographic transmission and enriches the methodology of information steganographic transmission.

## 7. Conclusions

In this paper, we propose a novel method for character-level text image steganography based on adversarial attacks. The way our information is embedded is very invisible, the carrier used and the key used to extract it (local model) have open uses. We make the OCR model steganographically capable. We exploit the vulnerability of artificial intelligence models and generate an adversarial attack that works only for the local model by limiting the transferability of the adversarial attack. We embed the steganographic information in Chinese text images which can be only extracted by the local model, while other OCR models have difficulty discovering the information. We show through decoupling experiments that our proposed adversarial transferability reducing method is effective and can limit the generated perturbations, and we achieve a good steganography success rate. We further explore the potential capabilities of our method by testing the performance of different hyperparameters on our steganography algorithm, and also verify the effectiveness of our validation model filtering strategy. The experiments show that our algorithm can obtain more than 97% steganographic success rate and 100% extraction success rate. In addition, We analyzed the performance of our algorithm in four aspects: capability, security, payload capacity, and robustness. Our algorithm has a good performance in steganographic capability and security, while the payload capacity and robustness of our algorithm may become a limitation of algorithm. In the future, we will focus our attention on improving the payload of our algorithms.

## Figures and Tables

**Figure 1 sensors-22-06497-f001:**
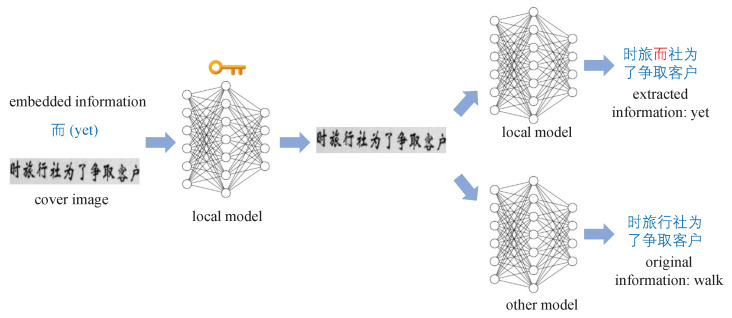
The schematic diagram of our steganography method. We achieve steganography by attacking the local model. We achieve steganographic message embedding by attacking the local model, and information extraction by capturing the character whose result recognized by the local model is different from the reference model.

**Figure 2 sensors-22-06497-f002:**
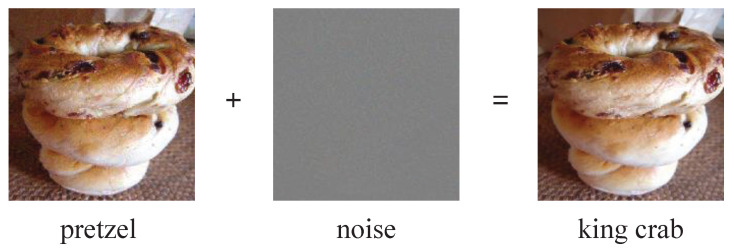
Flowchart of the generation of adversarial samples. After adding almost invisible noise to the image of a pretzel, the AI model adjudicates with a high degree of confidence as a king crab.

**Figure 3 sensors-22-06497-f003:**
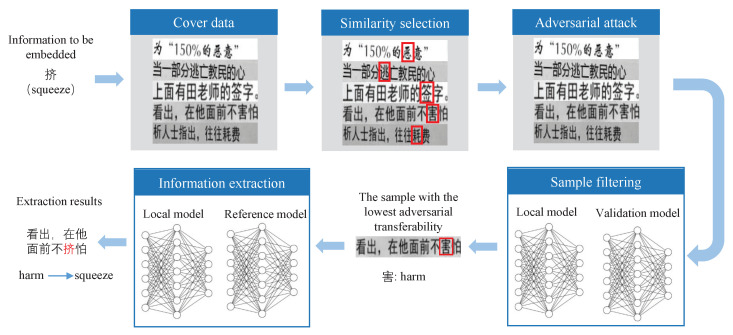
The framework of the character-level text image steganography method based on adversarial attacks. The red boxes are the characters we selected for modification. We have embedded the character squeeze in the character harm.

**Figure 4 sensors-22-06497-f004:**
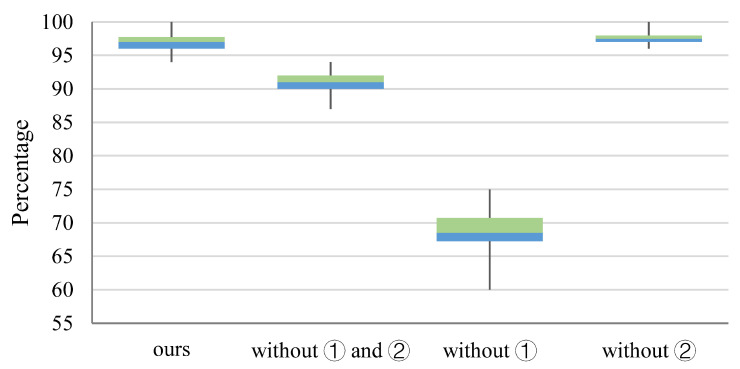
Statistical distribution of success rate of information extraction success rate.

**Figure 5 sensors-22-06497-f005:**
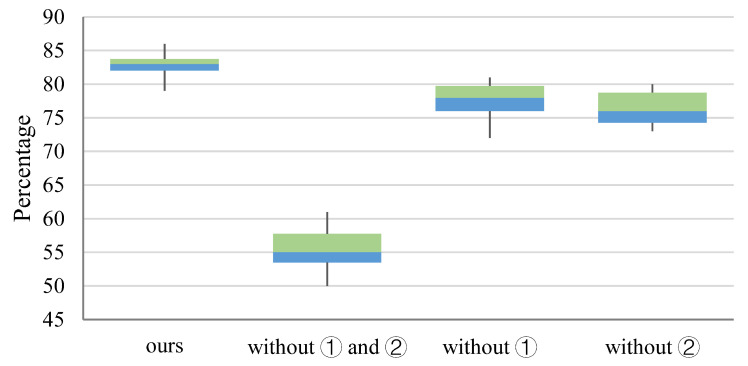
Statistical distribution of the probability of no impact on testing models’ results.

**Figure 6 sensors-22-06497-f006:**
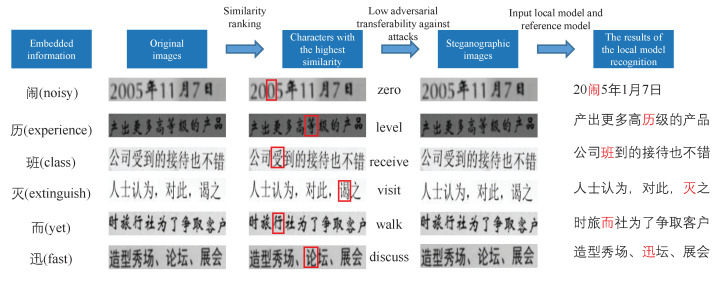
The steganographic image generated by our method. The semantics of the selected character in the original sample is completely changed.

**Table 1 sensors-22-06497-t001:** Average of the evaluation metrics we used.

Method	Extraction Success Rate	No Impact on the Result of the Testing Models	Correct Recognition Rate of the Testing Model	b Success Rate	$L_{2}$ Distortion
Ours	96.9%	82.8%	95.7%	80.23%	0.2137
Without ①	90.8%	55.6%	72.3%	50.48%	0.3239
Without ① and ②	68.2%	77.3%	89.2%	52.72%	0.2760
Without ②	97.6%	76.4%	88.3%	74.57%	0.2818

**Table 2 sensors-22-06497-t002:** Statistics of the variation of extraction success rate, the correct recognition rate of the testing model, probability of no effect on the testing model, steganography success rate, $L_{2}$ distortion with regular term coefficients.

Regular Term Coefficient	Extraction Success Rate	No Impact on the Result of Testing Model	Correct Recognition Rate of Testing Model	Steganography Success Rate	$L_{2}$ Distortion
10	99.8%	80.2%	92.1%	80.04%	0.2369
20	99.3%	81.4%	94.5%	80.83%	0.2295
30	98.2%	81.5%	94.4%	80.03%	0.2243
40	96.9%	82.8%	95.7%	80.23%	0.2223
50	95.8%	82.3%	93.6%	78.84%	0.2187
60	94.9%	81.6%	92.8%	77.44%	0.2159

**Table 3 sensors-22-06497-t003:** Statistics of the variation of steganographic success rate, correct recognition rate of the testing model, probability of no impact on the testing model, correct information extraction rate, and $L_{2}$ distortion with different number of samples per group.

Numbers of Samples per Group	Extraction Success Rate	No Impact on the Result of the Testing Model	Correct Recognition Rate of the Testing Model	Steganography Success Rate	$L_{2}$ Distortion
1	96.9%	82.8%	95.7%	80.23%	0.2223
2	99.20%	94.30%	98.60%	93.55%	0.2187
3	100%	95.90%	98.8%	95.90%	0.2254
4	100%	96.60%	98.90%	96.60%	0.2179
5	100%	97.20%	98.80%	97.20%	0.2214

**Table 4 sensors-22-06497-t004:** Confusion matrix of embedded and extracted information when only one sample per group.

	Extracted Information		
**Embedded Information**	**TRUE**	**FALSE**	**None**
positive	957	4	39
negative	0	0	9000

**Table 5 sensors-22-06497-t005:** Confusion matrix of SRNet’s detection results.

	Detection Results	
**Embedding Information**	**TRUE**	**FALSE**
postive	0	1000
negative	0	1000

**Table 6 sensors-22-06497-t006:** Effects of different compression qualities on steganographic information extraction.

q	100	90	80	70	60
extraction success rate	100%	73.7%	59.7%	48.4%	44.8%
local model recognition rate on others	100%	99.98%	99.95%	99.87%	99.62%
reference model recognition rate	100%	99.99%	99.94%	99.88%	99.63%

where q denotes the JPEG compression quality.

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
