# Peer review of "A Novel Steganography Method for Character-Level Text Image Based on Adversarial Attacks"

_sensors, 2022, doi:10.3390/s22176497_

Round 1
Reviewer 1 Report
The paper is well-written and easy to understand. However, I found that the motivation of the study is quite generic and the paper lacks good examples to show the relevance of the problem. I would like to see a section or subsection which focuses on the motivation for the whole paper specifically in a separate section.
The findings are shallow without clear insights and practical value. The results are briefly reported in the paper, and lack clear insights about the study.
Can you show a confusion matrix? It would be great to see false positives and negatives for a better assessment of the performance of the model.
Did you perform parameter hypertuning? If yes, please provide a table with the values to enable replicability of the study.
The similarities and differences between the related work and the proposed work is not clear. It is slightly mentioned by the end of Section 2. Please elaborate.
The limitation section is missing.
The implication section is missing. Please add scenarios that can help the researchers, practitioners, and educators, and tool builders benefit from the study?
Reviewer 2 Report
This paper proposes a novel method for character-level text image steganography based on adversarial attacks. But the validity of the proposed method has been verified only for Chinese characters.
The authors describe that character-level attacks on English OCR models are less difficult than those on Chinese OCR systems because English has fewer elements and a more similar structure; whereas Chinese characters have a very large number of constituent elements. This means that the propped method cannot be applied to English characters, right? The authors should clarify the reason in a more quantitative or qualitative manner. Otherwise, the proposed method is just a special one only for Chinese characters which means it is useless for most (non-Chinese) readers.
Reviewer 3 Report
The authors present a steganographic method for embedding secret information into scanned Chinese text images at character level using adversarial attacks.
The aim is to have an embedding and extraction process based on using and fooling neural networks such as to make the local model able to embed and extract but other models incapable of extracting the secret hidden message.
There are several aspects that need correction and the presentation as a whole has to be clearer. Furthermore the methodology needs to be refined and the results assessed for robustness, capacity, security against simple statistical analysis, to name just a few.
In the following there are some aspects (the list is not complete) to highlight several observations:
- the most popular image steganography methods are not necessarily embedding the secret message on the whole image - you mention this aspect several times
- encryption is the process of hiding the semantics of information through cryptography. In steganography we use "embedding" to hide the existence of the message
- in lines 30-31 there is a confusion of terms: "steganography algorithmically embeds the confidential data into the redundant space of the public carrier data to generate the cover data" --> the stego object is generated this way, not the cover object
- when describing the goal of steganography, one should not forget about the secure and efficient extraction of the original secret data - see lines 33-34
- it is not clear what you mean by "unlike ordinary images, text images contain more information" - line 41.
- the steganographic method is described as: "Our method does not have a specific mapping for embedding the specified information in either the spatial or the frequency domain; the exact embedding depends entirely on the carrier and the local model" - lines 56-58.
Does this mean for certain carrier objects the methodology chooses to use the spatial domain and for others the frequency domain? Or is it the choice of the user? Clarification is needed.
- The introduction with highlights the importance of steganography for secure communication. In this context, how does the sender and receiver arrive to the same local model - how do they train the network identically - to embed and extract the data? How do they communicate all necessary information and parameter sets, etc.
- for introducing Adversarial attacks and samples - a suggestive figure could be very useful
- in Section 2 you present the different approaches to deep learning image steganography. Consider a clearer contrast on how GANs and adversarial attacks are different, and carefully classify your approach - it is not clear what the uniqueness of neural network boundaries mean.
- how are perturbations generated? and how are the recognition errors captured - on the receiver side? Please clarify
- In section 4.1 in the process of finding similarities - is it correct to conclude that only one word is embedded - in the most similar (but not identical) word representation (character)? Can we say that the method is a concealed substitution of one character with another?
- The first paragraph of section 4.1.1 needs careful rephrasing in order to convey the purpose of preprocessing correctly. The embedding information is the steganographic key? How is the reference model generated?
- in line 217 you propose the expanding of the sample size of each group - there is no previous mention on what a group means. A clearer description is necessary
- in line 239 remove the redundancy of the term model
- in line 264 is in not clear what is meant by: "we embed the information for randomly selecting the top 2000 high-frequency Chinese characters"
Do you mean the secret message is randomly selected from the most popular characters?
- please rephrase - line 267: " When OCR models with the same structure are difficult to check the embedding information, OCR models with other structures are more difficult to identify"
- the metrics have to be defined clearly. What does steganography success rate mean? And so on...
- the experimental part may become more relevant once the aove observations are handled
- for analyzing the steganographic strength, we also need to consider the payload - and how the system responds to increasing volumes of secret data, the robustness of the system, other statistical methods of steganalysis.
- the Conclusions should also provide a glance on the limitations of the system and consider introducing a comparison to other methods in the literature.
Round 2
Reviewer 2 Report
No suggestions.